# Neonatal Screening for Cystic Fibrosis in Hungary—First-Year Experiences

**DOI:** 10.3390/ijns9030047

**Published:** 2023-08-23

**Authors:** Andrea Xue, István Lénárt, Judit Kincs, Hajnalka Szabó, Andrea Párniczky, István Balogh, Anna Deák, Péter Béla Monostori, Krisztina Hegedűs, Attila J. Szabó, Ildikó Szatmári

**Affiliations:** 1Paediatric Centre, MTA Centre of Excellence, Semmelweis University, 1083 Budapest, Hungary; 2Department of Paediatrics, Albert Szent-Györgyi Medical School, University of Szeged, 6725 Szeged, Hungary; 3Heim Pál National Paediatric Institute, 1089 Budapest, Hungary; 4Division of Clinical Genetics, Department of Laboratory Medicine, Faculty of Medicine, University of Debrecen, 4032 Debrecen, Hungary

**Keywords:** cystic fibrosis (CF), newborn screening (NBS), immunoreactive trypsinogen (IRT), pancreatitis-associated protein (PAP)

## Abstract

The aim of this study is to evaluate the strategy of the cystic fibrosis newborn screening (CFNBS) programme in Hungary based on the results of the first year of screening. A combined immunoreactive trypsinogen (IRT) and pancreatitis-associated protein (PAP) CFNBS protocol (IRT/IRT×PAP/IRT) was applied with an IRT-dependent safety net (SN). Out of 88,400 newborns, 256 were tested screen-positive. Fourteen cystic fibrosis (CF) and two cystic fibrosis-positive inconclusive diagnosis (CFSPID) cases were confirmed from the screen-positive cases, and two false-negative cases were diagnosed later. Based on the obtained results, a sensitivity of 88% and a positive predictive value (PPV) of 5.9% were calculated. Following the recognition of false-negative cases, the calculation method of the age-dependent cut-off was changed. In purely biochemical CFNBS protocols, a small protocol change, even after a short period, can have a significant positive impact on the performance. CFNBS should be monitored continuously in order to fine-tune the screening strategy and define the best local practices.

## 1. Introduction

Cystic fibrosis (CF) is a multi-system genetic disorder with autosomal recessive inheritance. Pathological variants of the cystic fibrosis transmembrane regulator (*CFTR*) gene result in defective chloride transport at the apical surface of the epithelial cells [1,2]. The requirement of CF-positive diagnosis is a positive sweat test and/or two CF-causing mutations: if the sweat chloride concentration is higher than 60 mmol/L and/or two pathogenic mutations are present in the trans position, the case is classified as CF [3,4]. Newborn screening (NBS) of CF together with appropriate preventive care and management provides an improved prognosis and opportunity for a better quality of life for affected persons [5]. As an efficient and cost-effective public health strategy, NBS for CF is increasingly adopted internationally. All CFNBS algorithms employ measurement of IRT from dried blood spots (DBS) as the first step, but subsequent steps in screening vary considerably across countries. There are two major strategies: second-tier DNA analysis or second-tier biochemical analysis of the first DBS sample, followed by a second DBS sample analysis. DNA-based protocols are used, for example, in Norway [6], Switzerland [7], and the USA [8]. Although IRT/DNA strategies show slightly better sensitivity and specificity, biochemical protocols are widely used due to legislative and ethical issues. Biochemical CFNBS is performed in Germany [9], Austria [10], and Andalusia [11], as examples. Among biochemical approaches, the IRT/IRT strategy shows a relatively low PPV [12]. As a consequence, several programmes determine pancreatitis-associated protein (PAP) levels as a second-tier parameter in the framework of an IRT/PAP/IRT strategy.

Originally, Sarles et al. described an IRT/PAP protocol with two IRT-dependent cut-offs [13]. Weidler et al. showed that the IRT×PAP product presents better discrimination for CF than PAP alone as a second-tier screening parameter [14]. Such protocols, where the initially measured IRT value is included in the second-tier evaluation and a second IRT value is assessed, can achieve sufficiently good sensitivity and PPV. An IRT-dependent safety net strategy [15] as a complement to the above strategy could further improve sensitivity by detecting cases with markedly high IRT but relatively low PAP values. Importantly, a sweat test is used to confirm the diagnosis of CF for all screen-positive babies, followed by the genetic analysis of the *CFTR* gene [3].

In some cases, the diagnostic requirements are not fully met [16], i.e., where the sweat chloride value proves to be intermediate (concentration is 30–59 mmol/L) and the DNA analysis identifies 1 or no known pathogenic variants, or the sweat test is normal (chloride concentration below 30 mmol/L) and two *CFTR* gene mutations are identified, at least one of unclear outcome. Such cases are regarded as cystic fibrosis screen-positive inconclusive diagnosis (CFSPID) in Europe or cystic fibrosis transmembrane conductance regulator-related metabolic syndrome (CRMS) in the USA [17,18]. Biochemical CFNBS protocols have the advantage of detecting fewer carriers and CFSPID newborns compared to the IRT/DNA approach [19]. In order to keep the number of missed CF cases as low as possible, several factors should be considered when starting a CFNBS programme: IRT cut-off values, laboratory practices, and strategies [20], which all depend on local conditions.

The objective of this work is to present the results and experiences of the first year of the CFNBS programme in Hungary preceded by a two-stage pilot study, to evaluate the analytical performance of the applied IRT/IRT×PAP/IRT+SN strategy of the current screening algorithm and to identify possibilities for improvements.

## 2. Materials and Methods

### 2.1. Pilot Study Design

A two-stage pilot project was applied. In the first stage, IRT values were determined in 10,000 anonymous newborn samples, and percentile (pc)-based cut-off values were calculated. In the second stage, following written informed parental consent, an additional 1719 newborn samples were assessed. Both studies were conducted at the Hungarian NBS Centres located at Semmelweis University, Budapest, and Albert Szent-Györgyi Medical School-University of Szeged. The studies were carried out in a prospective manner.

### 2.2. Samples and Exclusion Criteria

Dried blood spot samples were collected through the ongoing nationwide NBS system and analysed at the assigned NBS Centres in Semmelweis University, Budapest, and Albert Szent-Györgyi Medical School-University of Szeged. For IRT and PAP quantitation, the sampling was performed according to the general NBS sampling criteria, at the age of 48–72 h. The first DBS card (IRT1) was evaluated for CF screening only if the gestational age was at least 32 weeks. If samples were taken before the 48th hour of life or before the 32nd gestational week (GW), a second DBS sample was requested and assessed, irrespective of the result obtained from the first DBS sample. In case of delay (2 weeks from request date), in order to receive the second sample as soon as possible, the families were contacted by phone and health visitors were notified and involved when necessary.

### 2.3. Screening Test Methods

IRT concentrations from DBS samples were measured with the DELFIA Neonatal IRT Kit (PerkinElmer, Turku, Finland) and PAP levels were determined by the MucoPAP-F Kit (Dynabio, Marseille, France) according to the manufacturers’ instructions. Samples were initially assayed as a single determination. Samples with results above the cut-off were re-assayed in duplicate from the same card, but on a different spot to give a more definitive result. The mean of the measurements was considered for further decisions.

### 2.4. CF Diagnosis

Cases with positive screening results were referred to CF centres for sweat testing. Sweat chloride concentration was determined after pilocarpine iontophoresis with the Chlorochek^®^ Chloridometer using the Macroduct^®^ system for sweat collection (both EliTech Group Inc., Paris, France). The diagnosis of CF was set when sweat chloride concentration was above 60 mmol/L. When sweat chloride was less than 30 mmol/L, CF was excluded. In cases of borderline sweat chloride values (cc 31–59 mmol/L), the sweat test was repeated. For all cases with abnormal sweat tests (chloride concentration above 30 mmol/L), genetic investigation was initiated.

Genetic analysis was carried out in a centralised manner at the Division of Clinical Genetics, Department of Laboratory Medicine, Faculty of Medicine University of Debrecen, based on the 2018 ECFS best practice guideline [3]. Tests were carried out in accordance with the manufacturers’ instructions for use. In the first line, a *CFTR* mutation panel was used, which is able to detect at least one abnormal allele in >96% of the individuals in the Hungarian CF population [21]. The used Devyser *CFTR* Core kit (Stockholm, Sweden) is based on multiplex allele-specific PCR amplification and is suitable for detecting 36 mutations (711+1G>T, 3120+1G>A, 621+1G>T, 1717-1G>A, *CFTR*dele2,3(21 kb), 3849+10 kbC>T, 2789+5G>A, 1898+1G>A, G542X, G85E, Y1092X(C>A), G551D, R553X, 3659delC, N1303K, R560T, R117H, R1162X, L1077P, R117C, R1066C, L1065P, W1282X, R347H, R347P, I507del, T338I, F508del, I336K, 1677delTA, R334W, 3272-26A>G, 1078delT, 2183AA>G, 2184insA, 2143delT). The assay also detects poly-thymidine variants (5T/7T/9T) within intron 9. In the case of a 5T allele, the TG repeat number upstream of the poly-T region can also be determined. In cases when the newborn/infant had symptoms suggestive of CF and/or the first-line genetic test detected one pathogenic variant or one variant of varying clinical consequence, next-generation sequencing of the whole *CFTR* gene was performed. The used Devyser *CFTR* CE-IVD kit (Stockholm, Sweden) tests the coding region of the *CFTR* gene, as well as the exon–intron boundaries, the promoter region, and some clinically relevant deep-intron mutations. In addition to quantitative CNV detection, the direct detection of 8 common *CFTR* CNVs is also included. If no abnormal alleles were detected in the second line of testing, an additional MLPA (Salsa MLPA probemix P091 *CFTR*, MRC Holland, Amsterdam, The Netherlands) was performed to detect rarer CNVs.

### 2.5. Data Analysis

The data of both the pilot study and CFNBS were collected in NBS programme databases, and calculations were performed with Microsoft Excel.

## 3. Results

### 3.1. Pilot Study

Following multiple elaborate applications for the extension of the Hungarian NBS panel with CF screening, a two-stage pilot study was approved by the Medical Research Council Ethics Committee, Hungary, in 2017. Stage 1 was performed in 2017, and 10,000 anonymised DBS samples were included. The purpose of this stage was to calculate the IRT cut-off values for screening in the Hungarian newborn population. Based on the calculated percentiles—68.8 ng/mL for pc 95.0 and 119.8 ng/mL for pc 99.0—the cut-offs were defined as 65 ng/mL for positive cases and 120 ng/mL for SN. According to these results, a preliminary screening strategy was designed, as shown in Figure 1.

In the second stage of the pilot study (ethical approval codes 75/2019-SZTE and 104/2019-SE issued by the Medical Research Council Ethics Committee, Hungary), carried out in 2019, the previously designed strategy was put to proof. In total, 1719 samples were assessed, with written consent. Out of 18 IRT-positive (65–120 ng/mL) DBS cards, none showed to be positive for PAP. On the SN line, three babies were referred for clinical examination, and sweat chloride concentration was determined. Following genetic counselling and informed consent, the sequence analysis of the CFTR gene was performed. Based on sweat chloride concentration assessment and genetic testing, no CF cases were identified in the pilot study. Since 2019, both screening laboratories participate in the NBS Quality Assurance Program offered by CDC (https://nbs.dynamics365portals.us, accessed on 12 June 2023) and have met the requirements.

Due to ethical, legal, and economic concerns, a redesign of the preliminary protocol was requested. Consistent with the German and Austrian CFNBS protocols, a new screening strategy was designed (Figure 2). Cut-off values for IRT1 were kept at 65 ng/mL and 120 ng/mL (SN). The IRT×PAP product cut-off was set to 160 ng^2^/mL^2^ and IRT2 cut-off values were considered in an age-dependent manner (65 ng/mL up to 4 weeks of age; 50 ng/mL between 5th to 6th week of life; 30 ng/mL between 7th to 8th week of life). For cases older than 8 weeks, IRT-based screening was not applied [22].

### 3.2. CFNBS

In 2022, the CFNBS national programme finally started as part of the compulsory NBS in Hungary. A total of 88,400 samples were measured for IRT within the CFNBS program (Figure 2). Among them, 490 samples did not meet the sampling requirements (sampling age < 48 h or preterm babies < 32 weeks gestational age), and a second sample was requested. Of all accepted samples, 1331 proved to be positive for IRT1, calculated as an average of the measurements. The mean IRT values and the 95th percentiles were recorded each day that the assay was performed and plotted against time in order to assess the necessity of a floating cut-off (data not shown). Out of the 1331 positive samples, 1171 presented IRT values between 65 and 120 ng/mL, and 160 were above the SN cut-off, and immediately reported for clinical evaluation without further biochemical testing. The PAP values of the SN cases were also determined in order to evaluate the screening protocol for further adjustments. Out of the 160 SN cases, 155 had high IRT×PAP product.

Since for 65 cases, the second sample (IRT2) was already available prior to PAP measurements, only 1106 samples were to be tested for second-tier PAP values. According to the calculated IRT×PAP product values (>160 ng^2^/mL^2^), 360 new DBS samples were requested for IRT2 measurements. The average age for the second sample was 36 days. In 150 cases, the second DBS sample did not arrive within 8 weeks. These cases were considered screen-inconclusive and were referred to clinical overview. Finally, 96 samples proved to be above age-dependent cut-offs. These babies, together with the 160 newborns with SN IRT1 values, were considered screen-positive cases and referred to clinical evaluation.

Following sweat tests and genetic analysis of the CFTR gene, 14 CF cases and 2 CFSPID cases were identified (Table 1). Due to the lack in supply of sweat tests, caused by the new EU Medical Device Regulation (REGULATION (EU) 2017/745), the sweat chloride concentration could not be determined in several screen-positive cases. As a consequence, for these cases, only genetic analysis was carried out at the time. Where two CFTR variants with uncertain clinical consequences were detected, cases were taken into clinical follow-up. Two false-negative cases were found based on clinical symptoms (in one case, IRT2 did not reach the week-defined age-dependent cut-off, and in the second case, IRT×PAP product was below 160 ng^2^/mL^2^). Sample quality in these two cases was retrospectively inspected, and no ambiguities were found.

Among all IRT1 positives and diagnosed cases, the percentage of females was 58% and 67%, respectively. The average birth weight (BW) of screen-positive cases (3075 g) showed to be in the normal BW range (2500–4500 g). The BW of the identified CF cases was in a normal range, too. Using the IRT×PAP product cut-off as a second-tier CFNBS parameter, the number of requested second samples did not differ significantly as compared to the two-level PAP cut-off strategy (Figure 3).

For the Hungarian CFNBS programme, based on the first year’s results, the main performance indicators, a PPV of 5.9% and a sensitivity of 88%, were calculated (Table 2). Since clinically diagnosed cases of CF were reported to the national NBS laboratory irrespective of their screening test result, the calculated sensitivity reported here has high reliability.

## 4. Discussion

Many CFNBS screening programmes are available internationally, with marked variations in protocol designs. Prior to implementation, the pilot study of the Hungarian CFNBS was used to determine the cut-off values to be applied during screening and to evaluate the protocol design. The IRT percentile values determined in the pilot programme met the observed values in other European CFNBS programmes. The set value of 65 ng/mL for the IRT cut-off is similar to the cut-off values applied in most European countries.

The main goal of NBS programmes is to detect infants with a treatable disorder early in order to initiate treatment [23]. However, besides detecting the most—preferably all—cases possible, the minimisation of false-positive results is also an important aspect. The optimisation of a screening protocol is indispensable in order to achieve the best possible effectiveness of the programme in line with expected performance characteristics (sensitivity, specificity, PPV). In order to improve these characteristics, several factors like gender, age at specimen collection, birth weight, and seasonal variations should be considered.

Hungary has a typical continental climate with hot dry summers and mildly cold winters. As described by Kay et al. [24], the mean IRT values were observed to be higher in winter months than during the summer. Since no relevant pattern was observed, the use of floating cut-off values was considered unnecessary. Based on the results obtained from the first year of screening and with the experiences gained during this time period, we plan to adjust the cut-offs accordingly in the future.

Taking into consideration that in samples of premature newborns, the level of IRT is higher [25] and the level of PAP is lower [26] than the average of term babies, a new sample was requested in all cases where gestational age was less than 32 weeks. The same applied to cases where the sampling age was earlier than 48 h—the stress experienced during birth might increase IRT levels in newborns [27]. In accordance with this, 93% of DBS samples taken at an age younger than 48 h presented IRT values above 120 ng/mL, and 88% of all early (<48 h) samples ultimately proved to be screen-negative. Excluding these results (and asking for a second sample), the number of SN cases referred to clinical evaluation decreases significantly, and thus, the PPV% increases.

The majority of confirmed CF cases had an initial IRT value above 100 ng/mL. Only two confirmed CF positives, including one of the false-negative cases, fell into the 65–100 ng/mL range that accounted for 126 of the requested 360 s DBS cards. The IRT2 value of the false-negative case was 36 ng/mL. The sampling age for IRT2 had been calculated as 6 weeks, and according to the age-dependent cut-off values based on weeks (5th to 6th week of life, 50 ng/mL), the case was considered screen-negative. Following the symptom-based identification of the false-negative case, the age-dependent cut-off value definitions were modified to be based on days instead of on weeks, as follows: 65 ng/mL up to 28 days of age, 50 ng/mL between 29 and 42 days, and 30 ng/mL between 43 and 56 days. According to the updated definitions, the sampling age for IRT2 of the false-negative case was 43 days and the case could have been considered screen-positive.

Based on the genetic analysis of the *CFTR* gene, the F508del homozygotes were in the majority; however, no associations were observed between distinct mutations and the quantities of IRT and PAP parameters.

Out of the identified 14 CF and 2 CFSPID cases, only 3 CF cases and 1 CFSPID case showed an IRT1 value below the SN cut-off. Eleven CF cases and the other CFSPID case fell into the SN IRT1 range. Defining the age-dependent cut-off in days instead of weeks from the beginning of the regular CF screening would have allowed the identification of one false-negative case. On the other hand, had IRT category limits been defined as “less than 120” for PAP second-tier and “more than or equal to 120” for SN, this case could have been identified by the sweat test. The other false-negative case passed the CFNBS net irrespective of the strategy applied or cut-off set. This finding is in line with studies showing that false-negative cases are possible even with optimised CF screening approaches [28].

In the European Cystic Fibrosis Association Standards of Best Practice Guideline, it is stated that after CFNBS positivity observation, patients should be seen by a CF specialist team at a mean of 35 days and no later than 58 days [3]. All screen-positive cases were called in for clinical investigation earlier than 58 days of age. The age of diagnosis for screen-positive cases was between 32 and 100 days, and the false-negatives (FN1 and FN2) were identified at the age of 112 and 109 days, respectively. It also has to be noted that in the year 2022, due to the difficulties encountered because of the Medical Device Regulation for the European Union, Macroduct^®^ and ChloroChek^®^ Chloridometer kits (ELITech Group, Paris, France) were temporarily unavailable. Since the results of genetic testing were obtained in a longer time as compared to sweat test results, the final diagnosis was set at a higher age. The same guideline recommends that national screening programmes should aim for a PPV of at least 30% and a sensitivity of 95% [3]. Compared to an IRT/IRT/DNA protocol, an IRT/PAP/IRT protocol has two main weaknesses: it results in lower sensitivity compared to a well-functioning IRT/DNA protocol, and as a purely biochemical protocol, it has a very low positive predictive value. However, the combination of IRT/PAP with a second IRT as a third tier is an alternative for a sufficiently performing CFNBS protocol. Even if the recommended PPV is reported to be hardly achievable in purely biochemical protocols [19], we are striving to approach it as much as possible, while maintaining sufficient sensitivity.

With the changes in the decision chain applied following recognition of FN cases (i.e., week-based vs. day-based cut-offs, and “higher” vs. “higher or equals to”), the theoretic sensitivity would be 94%. Based on the obtained results, the incidence of CF in Hungary was calculated as 1/5200, in good accordance with data for other European populations [29].

## 5. Conclusions

The present study, describing the first year’s CFNBS results of the Hungarian screening, underlines the importance of NBS in the predominant route of diagnosing CF. As in most countries, the first year of the programme functioned with a relatively high false-positive rate. Even technical aspects (e.g., short supply of consumables caused by specific regulations) have a profound effect on statistical characteristics. As the implementation of a new programme generally encounters difficulties, it is highly probable that the effectiveness of the programme will improve in the following years. A small protocol change, even after a short period, can have a significant impact on the performance, as shown by an improved detection of CF cases using IRT2 cut-offs based on days instead of weeks. In any case, CFNBS should be monitored continuously, in order to fine-tune the screening strategy and to define best local practices aiming to increase the effectiveness of the programme.

## Figures and Tables

**Figure 1 IJNS-09-00047-f001:**
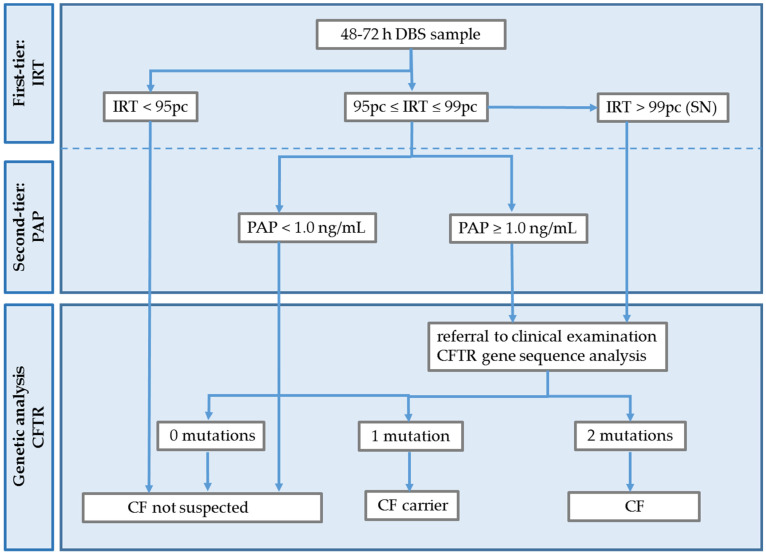
Preliminary CF screening strategy following pilot Stage 1, applied in pilot Stage 2.

**Figure 2 IJNS-09-00047-f002:**
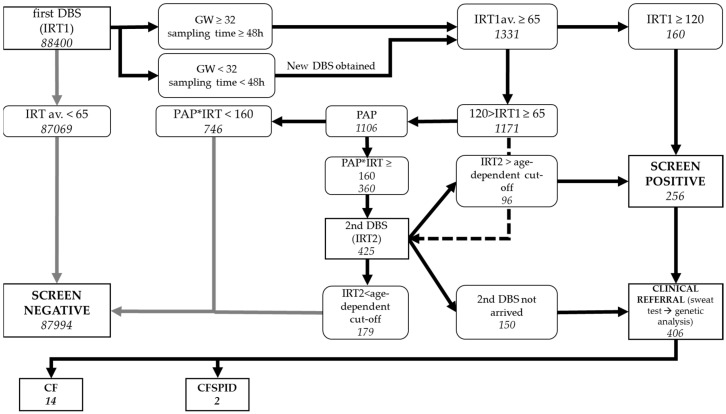
Final Hungarian CFNBS algorithm with number of samples within indicated categories.

**Figure 3 IJNS-09-00047-f003:**
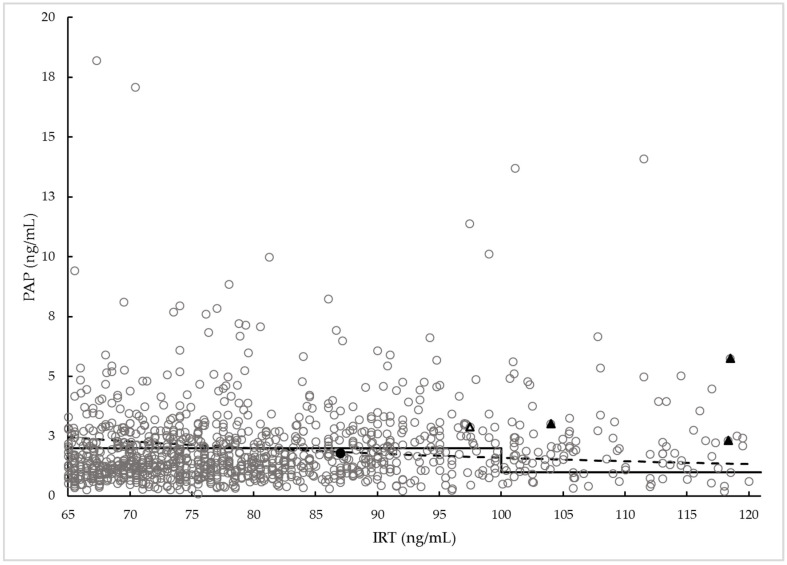
IRT and PAP concentration of CF (full triangle), CFSPID (open triangle), healthy, IRT1-positive newborns (open circle) and the false-negative cases (full circle). The dashed line represents IRT×PAP product cut-off; the solid line represents 2-level PAP cut-offs (for IRT1 between 65 and 100 ng/mL PAP 2 ng/mL, for IRT1 between 100 and 120 ng/mL PAP 1 ng/mL). For better visibility, cases exceeding the safety net limit (IRT > 120 ng/mL) are not depicted.

**Table 1 IJNS-09-00047-t001:** CFNBS and genetic results of CF and CFSPID cases identified in 2022. MI, meconium ileus; FN, false-negative; BW, birth weight; GW, gestational weeks. The values considered for clinical referral are typed in bold.

Patient ID	IRT1 (ng/mL)	PAP (ng/mL)	IRT×PAP (ng^2^/mL^2^)	IRT2 (ng/mL)	ST (mmol/L)	Mutation	GW (Weeks)	BW (g)	Gender	Age at Diagnosis (Days)
CF1(FN) *	121.0	-	-	36.0	-	F508del/F508del	38	2870	F	112
CF2(MI)	121.0	-	-	100.0	-	F508del/F508del	37	3250	F	intrauterine
CF3	104.0	3.0	365	**108.6**	Cl: 82	F508del/F508del	39	3090	F	85
CF4	118.5	5.8	713	**55.0**		F508del/F508del	40	4150	M	134
CF5	**143.5**	6.5	933	-	Cl: 71	F508del/F508del	41	3400	F	60
CF6	**140.0**	11.0	1540	-	Cl: 79	F508del/E92 *	40	3660	F	39
CF7(FN)	87.5	1.6	139			F508del/F508del	39	3000	M	109
CF8	**273.7**	37.6	10,290	-		F508del/2143delT	34	2140	F	53
CF9	**216.3**	5.3	1149	-	Cl: 96	F508del/F508del	38	2390	F	35
CF10	**359.0**	28.2	10,124	-		F508del/F508del	38	2900	M	48
CF11	**640.7**	24.1	15,440	-		F508del/F508del	39	2650	F	24
CF12	**283.0**	4.9	1384	-		F508del/2789+5G>A	39	2920	M	39
CF13	118.3	2.3	275	**106.5**	Cl: 80	F508del/F508del	38	2750	F	58
CF14	**342.3**	8.8	2999	-	Cl: 98	F508del/G745X	38	3870	F	37
CF15	**281.0**	15.4	4327	-	Cl: 35	F508del/F508del	39	3000	M	30
CF16 **	**143.0**	2.0	286	-	Cl: 50	3272-26A>G/C524X	36	3060	M	190
CFSPID1 ***	97.5	2.9	282	-	Cl: 52	F508del/5TTG12	39	2900	F	96
CFSPID2	215.0	4.5	968	-	Cl: 21	F508del/R117H;7T	40	3480	F	118

* Second sample was available prior to PAP measurement; ** due to borderline ST result, the diagnosis was set based on genetic test; *** second sample did not arrive.

**Table 2 IJNS-09-00047-t002:** Statistical characterisation of Hungarian CFNBS in the year 2022.

Screening for CF Characteristics	IRT/IRT×PAP/IRT+SN
Infants screened for CF	88,400
Infants with CF + CFSPID (% of infants screened)	16 + 2 (0.02%)
CF screen-positive results (% of infants screened)	256 (0.29%)
Infants referred to CF centre for sweat test	406
Detected cases with CF (% of infants with CF+CFSPID)	14 (78%)
False-positive cases (% of infants screened)	238 (0.27%)
Detected CF +CFSPID related to CF screen-positive cases	~1:16
CF screen-negative results (% of infants screened)	87,994 (99.5%)
True-negative cases (% of infants screened)	87,992 (99.5%)
False-negative results (% of infants with CF+CFSPID)	2 (11%)
Sensitivity	88%
Specificity	99.7%
PPV	5.9%

## Data Availability

All research data can be found in the Hungarian NBS database.

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
