# Peer review of "Neonatal Screening for Cystic Fibrosis in Hungary—First-Year Experiences"

_2409-515X, 2023, doi:10.3390/ijns9030047_

Round 1

Reviewer 1 Report

This Hungarian CF-NBS program is in alignment with other European programmes.

It is well written, well documented and the methods are solid.

Some ajustments have been made within the first year - improving the screening.

PPV is as expected with the method.

L 247: IRT2 at 6 weeks - could that be improved?

No further comments.

Reviewer 2 Report

This is a very important paper about the introduction of CF newborn screening in Hungary and should definitely be published in the journal. However, there is still a lot of work to be done before the status for a possible acceptance of the paper is reached. In detail, the following points should be addressed and the paper should be revised accordingly. 

Major points:

How was the percentile chosen? Was 65 ng/ml given, because that is the value in many countries, and then the percentile calculated? Or was the percentile from other countries given and then the IRT value determined? It is very surprising, but in our screening programme 65 ng/ml measured with Auto-Delfia corresponded to the 99.0 percentile and this is more or less the case in many European countries. 

In my opinion, both percentiles were chosen extremely low. What was the reason for this? If blood is drawn between the 48th and 72nd hour of life, an IRT cut-off based on the 99.0 percentile or even slightly higher should be sufficient for sufficient sensitivity. Lower percentiles are only necessary where very early blood draws are performed, e.g. in the USA. In Europe, where most NBS programmes currently do not collect blood before 36 hours of life, such a low IRT percentile is not necessary. It only worsens the PPV unnecessarily! 

It is not entirely clear from the paper what was measured in pilot study stage 1 and 2 and how. It is absolutely understandable that changes take place here, but it is not clear when the algorithm under Figure 1 was used and when the algorithm under Figure 2 was used. Can this be narrowed down in time and with a number of newborns screened for CF? It should be known that the population of Hungary is not so large that relevant numbers can be expected here. Nevertheless, it is absolutely important for transparency to get this information as a reader. If necessary, a table can be inserted for this purpose.  

Page 5, Figure 2: There are international rules for the use of symbols in flowcharts. Actually, it's not a big problem if you don't follow them, because simple flowcharts are usually understood anyway. But it gets complicated when suddenly symbols like a rhombus, which is definitely reserved for "decision points", is used somewhere as an end point. That is the case here and that is why my suggestion would be that with this complexity of the flowchart, one should stick to the international rules! 

Page 5, Line 172: „Since for 65 cases the second sample (IRT2) was already available prior to PAP meas-172 urements, only 1106 samples were to be tested for second-tier PAP values.“

If you want to evaluate a protocol, this is a mistake. In the context of pilot studies, it may happen that the steps do not go exactly according to the protocol, for various reasons. However, it does not make sense to limit the possibility of a meaningful subsequent evaluation by not carrying out these steps. It is not very scientific to assume that IRT/IRT is the gold standard before the possibly less reliable PAP measurement. The question then arises as to why it is evaluated at all. 

Page 6, Table 1: It would have been helpful to show with which protocol step the CF patients were finally found. Since PAP was obviously always determined except for the first two CF patients and the CF children were actually also found with "IRTxPAP", the question arises why the Safety Net is needed at all? The charming thing about the product "IRTxPAP" is that the Safety Net (SN) is practically already built in. If, for example, PAP had been determined in the first false-negative (FN) patient who was positive for SN, a value of 1.33 would only have been needed to reach the cut-off value of 160 ng2/ml2. This PAP value would have been negative for both the cut-off values (for MucoPAP-F) in the Netherlands (1.6 ng/ml) and in Germany (2.1 ng/ml), but would still have indicated a possible positive result due to the value of the product. If the IRT2 had been carried out at short intervals afterwards, or if this had not been successful, the parents would have been called in for confirmation diagnostics, the child would have been found!

The authors should go through their data and calculate whether this Safety Net is even needed in this form? They should state how many additional false positively detected newborns have occurred and they should state how many CF patients would not have been detected without the SN! 

Page 7, line: 209: „For the Hungarian CFNBS programme, based on the first year’s results, the main  performance indicators, a PPV of 5.9% and a sensitivity of 88% were calculated (Table 2).“

The achieved PPV is clearly too low in view of a target PPV of 30% according to the ECFS standard. It is known that IRT/PAP+SN algorithms do not achieve this. However, it is also known that this disadvantage can be improved with a third protocol tier. This is actually available with a 2nd IRT. The reason for the poor specificity and the poor PPV lies in the safety net (SN), which in my opinion is incorrectly placed in the algorithm and which, if fulfilled, immediately leads to confirmation diagnostics. This was nonsensically also used in the IRT/PAP/DNA protocol of Germany, which also leads to an unnecessary drop in the PPV there; unnecessary because the gain in sensitivity thus achieved is far too small. The authors should explain again why this was done.

Page 8, line 248-253: „Following the symptom-based identification of the false-negative case, the age-dependent cut-off value definitions were modified to be based on days instead of on weeks, as follows: up to 28 days of age 65 ng/mL, between 29-42 days 50 ng/mL and between 43-56 days 30 ng/mL. According to the updated definitions the sampling age for IRT2 of the false-negative case was 43 days and the case could have been considered screen-positive.“

Whether this was really a useful move remains to be proven. It is quite possible that this will result in more false-positive cases again, even if not many! However, with the first false-negative case from Table 2, there are still things that need to be questioned.

1. it makes absolutely no sense to allow step 3 to be done before step 2 and then to do without step 2 if the protocol is IRT/IRTxPAP/IRT. The determination of PAP as step 2 would have resulted in the chance that the child would have been detected positive here after all (see above).

2. it is probably generally better to accept only a limited window for IRT2 (e.g. only from 21 to 42 days or even less) and to immediately arrange for confirmation diagnostics in those who have not had IRT2 determined by then.

3) How can it actually be that this is a false-negative case at all? The child had an IRT1 of 121 ng/ml and the threshold value for the safety net was 120 ng/ml. As I understand it, the child would have had to have a confirmation diagnosis anyway, even though I do not consider this type of safety net in the procedure to be very appropriate otherwise!

 Page 8, line 281: „Since the problem of a too low PPV in purely biochemical protocols is not relevant [19], only the requirement of sufficient sensitivity remains valid.“

This statement is only true if I see the detection of carriers as the only problem. What is written here is true, but at least as big a problem is the triggering of fear and moments of shock in the parents of CF-positive tested children. There is a difference between accepting this for one diagnosed CF child with three or four additional healthy people found, and almost 20, as in this case. Also, sweat tests don't always work the first time and then the problem gets worse. Detecting too many false positives is also an economic problem if material and human resources are wasted. 

Minor points:

Page 6, Table 1: MucoPAP or MucoPAP-F? That really makes a difference for the discussion of the cut-off value! Or was it even measured with both kits? In the table, it would certainly be sufficient to write only "PAP" if there is clarity about the kit used in the method section.

Page 7, line 237: „The same applied 236 for cases where sampling age was earlier than 36 hours ...“ Before that, the text of the paper and Figure 2 speak of "<48 h"!

Reviewer 3 Report

This manuscript outlines the initial experiences of the Hungarian CF newborn screening protocol, highlighting the lessons learned from the initial pilot studies by reporting the program's current performance.   Due to the availability of genetic testing for CF in their region and in alignment with many European countries, the Hungarian program utilizes PAP to improve the screen's performance. The IRT/PAP/IRT algorithm described may be less timely than other models in use worldwide, but it does decrease the likelihood of CRMS/CFSPID diagnoses, as highlighted by the authors.

1)     Results:   Section 3.1: "Following multiple circuitous applications…" – Replace circuitous with iterative.   Ciecuitous can imply not being straightforward and can sometimes be seen as negative. 

2)     The IRT cutoffs proposed may be higher than seen in many programs in the US, although in alignment with European cutoffs, and IRT is known to shift over time, with lot variation and seasonal impacts. The program may wish to consider a floating cutoff.   

3)     The figures are very difficult to read.    They need to be submitted with better resolution.   They also seem to have been done in different computer programs or formats; consider making them consistent. 

4)     In Figure 1, consider putting in an additional rectangle in PAP section or additional verbiage to reflect that the PAP is tested on the same sample.

5)     The figure that was not shown would be interesting to see. It would be reassuring for the reader to see IRT over time in the Hungarian population, especially since the authors conclude that a floating cutoff is unnecessary.

6)     Eight weeks seems like a long time to wait for the second specimen to arrive.    Has the program considered a shorter window to refer to clinical evaluation?

7)     What is the typical time elapsed before getting a second sample?   

8)     How many of the infants referred to clinical evaluation reported for sweat tests?   Does this vary based on why they were referred (only one specimen submitted vs. having a presumptive positive test?  

9)     What is the system in Hungary to identify false negatives?   It can sometimes be challenging for the NBS programs to know about all the false negatives that occur, limiting data-based decisions for IRT cutoffs. 

10)  Please put the age at the time of diagnosis in Table 1.

11)  In the case of Patient 1, the first false negative, why was PAP not performed, even though the second sample had arrived? Would that have changed the decision related to that child?  

12)  There appear to be results presented for the first time in the discussion.    Please ensure that all results are first presented in the results section and only discussed in the conclusion/discussion sections.    

Round 2

Reviewer 2 Report

The revised manuscript can now be published in the present form. The answers and corrections sufficient!